# Bile Acids Induce Neurite Outgrowth in Nsc-34 Cells via TGR5 and a Distinct Transcriptional Profile

**DOI:** 10.3390/ph16020174

**Published:** 2023-01-24

**Authors:** Hayley D. Ackerman, Glenn S. Gerhard

**Affiliations:** 1Lewis Katz School of Medicine, Temple University, Philadelphia, PA 19140, USA; 2Department of Molecular Oncology, H. Lee Moffitt Cancer Center and Research Institute, Tampa, FL 33612, USA

**Keywords:** bile acids, FXR, TGR5, neurite outgrowth

## Abstract

Increasing evidence supports a neuroprotective role for bile acids in major neurodegenerative disorders. We studied major human bile acids as signaling molecules for their two cellular receptors, farnesoid X receptor (FXR or NR1H4) and G protein-coupled bile acid receptor 1 (GPBAR1 or TGR5), as potential neurotrophic agents. Using quantitative image analysis, we found that 20 μM deoxycholic acid (DCA) could induce neurite outgrowth in NSC-34 cells that was comparable to the neurotrophic effects of the culture control 1 μM retinoic acid (RA), with lesser effects observed for chenodexoycholic acid (CDCA) at 20 μM, and similar though less robust neurite outgrowth in SH-SY5Y cells. Using chemical agonists and antagonists of FXR, LXR, and TGR5, we found that TGR5 agonism was comparable to DCA stimulation and stronger than RA, and that neither FXR nor liver X receptor (LXR) inhibition could block bile acid-induced neurite growth. RNA sequencing identified a core set of genes whose expression was regulated by DCA, CDCA, and RA. Our data suggest that bile acid signaling through TGR5 may be a targetable pathway to stimulate neurite outgrowth.

## 1. Introduction

Neuronal loss in the brain or spinal cord is associated with a variety of neurodegenerative disorders, and has prompted pharmaceutical strategies aimed at preventing neuronal cell death and promoting regeneration. Of the many biochemical pathways potentially modulating neurodegeneration, bile acids have recently emerged as potential neuroprotective agents in a diverse group of diseases [1,2] including Parkinson’s disease, characterized by dopaminergic neuronal degeneration in the substantia nigra, Alzheimer’s disease, characterized by neuronal degeneration in the basal forebrain, Huntington’s disease, characterized by selective neuronal loss in the striatum, amyotrophic lateral sclerosis (ALS), characterized by loss of corticospinal and spinomuscular motor neurons, and spinal cord injury [3]. A growing body of in vitro and in vivo evidence supports a neuroprotective role for bile acids in major neurodegenerative disorders including ALS [3,4,5,6], with beneficial clinical trial effects reported for sodium phenylbutyrate-taurursodiol [7,8], which has recently been approved by the Food and Drug Administration (FDA), though its mechanism of action is not known [9].

Most prior work on bile acids as pharmaceutical agents has focused on those found in the bile of certain species of bears due to historical antecedents of bear bile as a therapeutic in China [10]. In particular, the bile acid tauroursodeoxycholic acid (TUDCA) is thought to confer neuroprotective effects through anti-apoptotic mechanisms, including inhibition of the extrinsic apoptotic pathway through inhibition of cell death receptors and caspase-3, reducing ER-mediated stress, and decreasing calcium efflux from the ER and caspase-12 activity, as well as inhibition of the intrinsic mitochondrial apoptotic pathway that reduces reactive oxygen species (ROS) production and inhibits Bax translocation and cytochrome *c* release [11]. In addition, TUDCA has been reported to inhibit and directly modulate pro-survival signaling pathways, and regulation of the expression of genes involved in cell cycle and apoptotic pathways [11]. However, bile acids have long been known primarily for their physicochemical roles in the absorption of lipids, and as signaling molecules for the farnesoid X receptor (FXR or NR1H4) [12] and the cellular receptor G protein-coupled bile acid receptor 1 (GPBAR1 or TGR5) [13,14]. FXR also heterodimerizes with RXR (retinoid X receptors) [15], whose ligands include retinoic acid, and both isoforms of LXR, LXRα (NR1H3) and LXRβ (NR1H2) [16], which serve as transcription factors to regulate gene expression by binding to DNA at specific sequences in the promoters and enhancers of target genes.

Because FXR can heterodimerize with RXR, and RXR with LXR, and induction of neurite outgrowth by retinoic acid is a well-studied phenomenon, we hypothesized that bile acids could also induce neurite outgrowth in cell line models capable of neurite growth induction by retinoic acid. Rather than TUDCA, we conducted experiments with physiological levels of several of the main bile acids that we observed to circulate in humans [17], and not the toxicological levels used in many studies [18,19]. We found that the bile acids deoxycholic acid (DCA) and chenodexoycholic acid (CDCA) could function as neurotrophic factors that can induce neurite outgrowth in a manner comparable to retinoic acid (RA) in mouse NSC-34 cells, considered to be a highly stable and widely used murine motor neuron cell line model [20,21], and in SH-SY5Y cells, a human neuronal cell model [22]. These results are consistent with the reported neurotrophic properties of TUDCA in an ALS motor neuron survival assay [23]. Using receptor agonists and antagonists, we found that bile acids appear to signal through TGR5 to induce neurite growth. We also identified several RA and bile acid responsive genes that were induced or suppressed during the initial phase of neurite induction by these compounds. These data provide an alternative mechanism for the long-observed anti-neurodegenerative effects of bile acids.

## 2. Results

### 2.1. Bile Acids Induce Neurite Outgrowth Comparable to RA

We conducted experiments using mouse NSC-34 cells [24], considered to be a highly stable and widely used murine motor neuron cell line model [20,21], and SH-SY5Y cells, a human neuronal cell model [22] in which retinoic acid is used to induce neurite outgrowth and functional characteristics of motor neurons. Neurite outgrowth was measured after exposure to basal differentiation medium (DM), 1 μM retinoic acid (RA), and 20 μM of the bile acid deoxycholic acid (DCA) for 6 days. NeuronCyto II, a Matlab-based software package that was developed to address the “crossover” issues in quantifying neuronal morphology [25,26], was used to measure neurite outgrowth. We found that 20 μM deoxycholic acid (DCA) could induce neurite outgrowth in NSC-34 cells that was comparable to the neurotrophic effects of 1 μM retinoic acid (RA) (Figure 1A). Similar, but slightly less robust effects, were observed for CDCA at 20 μM (Figure 2D and Figure 3). In SH-SY5Y cells, 1 μM RA and 20 μM DCA induced a less robust neurite induction, although optimum culture conditions and seeding densities were not established (Figure 1B).

### 2.2. FXR or LXR Stimulation Much Less Neurotrophic than RA

To determine if a known bile acid receptor was part of the mechanism of stimulated neurite outgrowth, we then used several available chemical agonists and antagonists of FXR, LXR, and TGR5 to determine whether they could act as a substitute for or block the effects of bile acids (Figure 2 and Figure 3). In addition to 1 μM RA, which served as the reference control, 20 μM DCA and 20 μM CDCA were used. The commonly used FXR agonist GW 4064 [27] at 1 μM and 10 μM, WAY 252623, a potent LXR agonist that has been reported to stimulate the proliferation of mouse neural progenitor cells [28] at 1 μM and 10 μM, the dual FXR/TGR5 ligand obeticholic acid (INT 747) [15] at 1 μM and 10 μM, and the TGR5 agonist RG-239, which is a derivative of the naturally occurring triterpenoid betulinic acid [29] at 1 μM and 10 μM, were used. To block bile acid pathways, Z-Guggulsterone, an inhibitor of FXR transactivation [30] at 5 μM, and LXRα antagonist GSK2033 [31] at 5 μM were used. As described for Figure 1, NSC-34 cells were incubated for 6 days, followed by imaging and processing of 40 fields per condition with NeuronCyto II.

The RA at 1 uM (Figure 2B and Figure 3) had the expected effect of strongly inducing neurite outgrowth, with a median neurite length of 25.3 μm in RA treated cells compared to 3.2 μm in DM treated control cells without the addition of RA (*p* < 0.001 using a two-tailed Mann–Whitney test assuming a non-Gaussian distribution after post hoc adjustment for multiple comparisons). DCA (Figure 2C and Figure 3) and CDCA (Figure 2D and Figure 3) at 20 μM both induced neurite outgrowth with DCA more potent (median length 38.4 μm) than CDCA (median length 20.5 μm). The median neurite length in DCA treated cells was significantly longer (*p* < 0.01 after post hoc adjustment for multiple comparisons) than in RA treated cells, while CDCA median length was not different from that in RA treated cells (20.5 μm vs. 25.3 μm). DCA thus appears to have more robust neurotrophic effects than RA.

The median length in cells treated with 1 μM GW4064 (Figure 2E and Figure 3), an FXR agonist, was 10.9 μm; this is significantly shorter than RA (*p* < 0.001 after post hoc adjustment for multiple comparisons), while 10 μM was toxic and resulted in cell death (Figure 2F). The LXR activator, WAY 252623, induced mean neurite lengths of 10.3 μm and 8.4 μm at 1 μM and 10 μM concentrations (Figure 2G,H and Figure 3), respectively, which is also significantly lower than RA (*p* < 0.001 after post hoc adjustment for multiple comparisons). FXR or LXR stimulation was much less neurotrophic than RA, DCA, or CDCA.

### 2.3. TGR5 Agonism Is More Neurotrophic than RA

Neurite length induced by 1 μM of the FXR/TGR5 dual ligand obeticholic acid (INT 747) was 15.6 μm (Figure 2I and Figure 3), significantly lower than induced by RA (*p* < 0.001 after post hoc adjustment for multiple comparisons), but at 10 μM was 22.3 μm (Figure 2J and Figure 3) and not statistically different than induced by RA. The dual specificity INT 747 is a much stronger ligand for FXR, thus the dose response may be due to increasing TGR5 agonist activity. Consistent with these results, the TGR5 agonist RG-239 at 1 μM induced a neurite length of 9.5 μm (Figure 2K and Figure 3), significantly lower than induced by RA (*p* < 0.001 after post hoc adjustment for multiple comparisons), but at 10 μM the median neurite length was 37.0 μm (Figure 2L and Figure 3), almost identical to that induced by DCA. TGR5 agonism thus appeared to be comparable to DCA stimulation and stronger than RA.

### 2.4. Neither FXR or LXR Inhibition Blocks Bile Acid-Induced Neurite Growth

FXR and LXR antagonists were then tested to determine if they could block the effects of DCA. Neither 5 μM of FXR antagonist Z-Guggulsterone (Figure 2M) nor 5 μM of pan-LXR antagonist GSK2033 (Figure 2O) had any effect on median neurite length by themselves, nor had any effect on DCA -induced (Figure 2N,P) median neurite length (38.2 μm with Z-Guggulsterone and 42.0 μm with GSK2033 vs. 38.4 μm without either). Thus, neither FXR nor LXRα antagonism could block bile acid-induced neurite outgrowth. These data suggest that TGR5 is a mechanism through which bile acids mediate their neurotrophic effects in NSC-34 cells.

### 2.5. RNA-Seq Identified Bile Acid Specific Genes

We sought to characterize the early initiating molecular events in neurite induction by bile acids using RNA-seq analysis [32] of NSC-34 cells. To identify early effects on gene expression induced by either RA or the bile acids DCA and CDCA, RNA was isolated from NSC-34 cells treated for 48 h, a time frame before significant morphological effects can be seen and presumably when early transcriptional responses are occurring to enable subsequent neurite outgrowth. Principal component analysis grouped DM (control) separately from DCA and CDCA (Appendix A), and log2 fold changes for transcripts plotted versus the mean of normalized counts over all genes showed more dispersion with CDCA than with RA or DCA (Appendix A).

Treatment with 1 μM RA identified 112 genes (Appendix A) whose change in expression level relative to control basal neurite induction media lacking RA met a nominal (*p* < 0.05) level of statistical significance. Three genes achieved a statistically significant adjusted *p*-value threshold, including Cyp26b1, which is the form of CYP26 that is expressed in the brain, and for which RA is not only a specific substrate but also a potent inducer of expression [33]. RA is also known to induce the other two genes, DKK3 [34] and CTSB [35].

Treatment with 20 μM DCA identified 44 genes whose change in expression level relative to control basal neurite induction media lacking DCA or RA met a nominal (*p* < 0.05) level of statistical significance (Appendix A). Treatment with 20 μM CDCA identified 94 genes whose change in expression level relative to control basal neurite induction media lacking CDCA or RA met a nominal (*p* < 0.05) level of statistical significance (Appendix A). Comparing these three datasets identified genes whose differential expression was similar in two or more of the treatment conditions. For RA and either DCA or CDCA, nine genes were found whose direction of differential expression was the same and whose magnitude of differential expression was highly similar (Table 1). Few data are available on the role of RA or bile acids in the regulation of these genes, though KIF2C [36], LGALS1 [37], ADNP [38], MARCKSL1 [39], and MKI67 [40] are known to play roles in neuronal functioning, including regeneration and differentiation. For DCA and CDCA genes, four genes were found whose differential expression was in the same direction and whose magnitude was highly similar (Table 2). Few data are available on the role of RA or bile acids in the regulation of these four genes, though RPS12 [41] and TPRG1 [42] are known to play roles in neuronal functioning.

## 3. Discussion

Bile acids are a heterogeneous group of structurally related molecules that are derived from the two primary bile acids produced in the liver, cholic acid (CA) and chenodeoxycholic acid (CDCA). CA and CDCA can be structurally modified in the liver to generate other bile acid derivatives; this includes undergoing conjugation reactions to generate TCDCA. Additionally, in the intestine, microbial dehydroxylation produces deoxycholic acid (DCA) that enters the enterohepatic circulation. However, due to the historical precedent set by the use of bear bile in ancient Chinese medicine [10], many studies have focused on the bile acid TUDCA.

In contrast to the primary biochemical function of bile acids to serve as emulsifiers of dietary lipids to facilitate intestinal absorption, the pathological roles of bile acids in the central nervous system have not yet been well defined. The acidic pathway of bile acid biosynthesis from cholesterol in the CNS appears to be dysregulated in ALS [43]. Increased levels of CSF bile acids [44] as well as non-esterified cholesterol and decreased serum levels of 26-hydroxycholesterol [45] have been found in ALS patients. Bile acid metabolism in ALS has also been implicated by genetic data [46,47]. A screen of a library of bioactive chemical compounds using a survival assay for mouse motor neurons carrying a human superoxide dismutase 1 (hSOD1) G93A transgene, identified TUDCA as a neuroprotective molecule due to its “strong neurite outgrowth-promoting effects”; this was corroborated in vivo by a statistically significant increase in NMJ innervation after three weeks of subcutaneous TUDCA injections into hSOD1^G93A^ transgenic mice relative to vehicle treated controls [23]. Interestingly, the authors tested 10 bile acids for neurite outgrowth but did not include DCA or CDCA. TUDCA has also been reported to decrease axon degeneration caused by H_2_O_2_ in primary cortical neurons of mice, and decrease tissue damage with oral TUDCA supplementation in a mouse model of spinal cord injury [48].

We found that DCA had more robust neurotrophic effects than RA. This was surprising given the amount of previous work on RA as a neurotrophic factor [49] and a developmentally important molecule. The stronger neurotrophic effects of DCA relative to CDCA are consistent with our findings that TGR5 signaling appears to mediate neurite outgrowth. DCA is a known activator of TGR5 [50,51], as is CDCA [52,53], though DCA is a more potent agonist of TGR5 than CDCA, which is reversed, i.e., CDCA is greater than DCA, for FXR agonism [54]. Of relevance to ALS, TCDCA is able to bind to the TGR5 receptor and activate it [55,56]. However, few data are available on the role of TGR5 in the nervous system. TGR5 is expressed in inhibitory motor neurons of the enteric nervous system [57] in both rodent and human brain tissue, and localizes to astrocytes and neurons where it appears to be activated by endogenous neurosteroids [58]. Activation of TGR5 has been associated with differentiation of several other cell types, including osteoblasts [59], cholangiocytes [60], and monocytes [61].

The early signaling events after TGR5 stimulation may be reflected in our gene expression data. However, TGR5 mediated signaling may have induced secondary and tertiary effects on gene expression over the time frame used for the experiments. Several of the genes dysregulated by bile acids have previously been implicated in neurodegeneration. For example, we found down regulation of LGALS1 with neurotrophic treatment with RA and DCA. Consistent with these results, levels of serum galectin-1, encoded by the LGALS1 gene, were significantly elevated, the mRNA expression level of LGALS1 was significantly increased and the promoter of LGALS1 was hypomethylated in ALS patients [62]. We also found increased expression of ADNP (activity-dependent neuroprotective protein) with RA and DCA treatment. NAP (davunetide), an eight amino acid peptide derived from ADNP, prolonged longevity, protected spinal cord neurons, reduced hyperphosphorylation of brain tau, and increased survival in ALS SOD1-G93A mice [63].

Two of the four DCA and CDCA-regulated genes have potential relevance to the nervous system. The synaptic vesicle (SV)-associated protein Mover/TPRGL/SVAP30 has been reported to downregulate proteasomal and autophagic degradation of components of synapses, helping to maintain their assembly and stability [42]. The Gas2l3 (growth arrest-specific 2-like 3) gene codes for a cell cycle protein that binds cytoskeletal actin filaments and microtubule networks. Knockdown of Gas2l3 expression in zebrafish embryos caused abnormal ventricle formation and brain dysmorphogenesis [64] The expression of another DCA/CDCA regulated gene, IKBIP, showed positive correlation with glioma grade [64].

The loss of motor neurons in amyotrophic lateral sclerosis (ALS), a neurodegenerative disorder characterized by atrophy of skeletal muscles, results in progressive weakness, and death within 2 to 5 years of diagnosis in most patients [65]. The cellular models of neurite extension we used are potentially relevant to ALS disease models. For example, NSC-34 cells transfected with a transactive response DNA-binding protein 43 (TDP-43) human ALS mutation M337V had a reduction in the number and length of neurite processes [66]. Similar observations were seen with NSC-34 cells and primary rat forebrain neurons transfected to express TDP-43 fragments [67]. The number and length of neurites were also decreased in NSC-34 cells transfected with the human SOD1 gene carrying the ALS G93A mutation [68]. In human induced pluripotent stem cell lines from patients with mutations in the FUS RNA binding protein (FUS) that were analyzed using their corresponding CRISPR-Cas9 gene-edited isogenic control lines, the ALS-causing FUS mutations resulted in decreased neurite outgrowth and decreased neurite regrowth following axotomy [69]. Similar results were also seen with motor neurons derived from mouse induced pluripotent stem cells from transgenic mice carrying the human ALS SOD1 G93A mutation [70].

Neurotrophic factors have been studied as potential therapeutics, although issues related to delivery, dosing/timing, and bioavailability have hampered clinical application [71]. Mimetics with more favorable bioavailability profiles have therefore been pursued [72], alongside the investigation of pro-neurogenic small molecules such as retinoids [73] which play key roles in neural differentiation, motor axon outgrowth, and nerve regeneration [74]. In a recent report on a randomized, placebo-controlled, phase 2 trial in ALS (CENTAUR), orally administered sodium phenylbutyrate-taurursodiol (PB-TURSO) over 24 weeks resulted in slower functional decline than placebo; this was measured by the ALSFRS-R score [7]. In an open-label extension of the CENTAUR trial, a 6.5 month longer median survival relative to those receiving placebo was found with initiation of PB-TURSO treatment at baseline [8]. The FDA has granted full approval of PB-TURSO for the treatment of ALS [9].

## 4. Materials and Methods

### 4.1. Cells

NSC-34 cells (Cedarlane catalog number CLU140) were used as a motor neuron cell line model [20,21]. Cells were maintained in Dulbecco’s modified Eagle’s medium (DMEM) supplemented with 10% fetal calf serum (FCS) [20] and subcultured every 2–3 days. For neurite outgrowth experiments, cells were seeded onto collagen I coated plates. The day after plating, the maintenance medium was replaced with a differentiation medium composed of 1:1 DMEM/F-12 (Ham), 1% FCS, and 1% modified Eagle’s medium nonessential amino acids. A measure of 1 uM all-trans retinoic acid (RA, Sigma-Aldrich) diluted in dimethyl sulfoxide (DMSO) was used as the standard positive neurite induction control, and various concentrations/combinations of bile acids, inhibitors, or activators were used as described. Media replete with RA or bile acids was replaced every 2 days for a total of 8 days.

Human neuroblastoma SH-SY5Y cells (Sigma, catalog number 94030304) were also used as a neurite model [22]. Cells were maintained in DMEM containing 10% fetal bovine serum (FBS). For experiments, cells were exposed to the differentiation medium (3% FBS in DMEM) with 1 μM all-trans retinoic acid (RA, Sigma-Aldrich) diluted in dimethyl sulfoxide (DMSO) as the standard positive neurite induction control, alongside which various concentrations/combinations of bile acids, inhibitors, or activators were used, as described, for 8 days. Media replete with RA or bile acids, inhibitors, or activators was replaced every 2 days.

### 4.2. Quantitative Image Analysis

Cell cultures were fixed with paraformaldehyde then immunostained with anti-α-tubulin antibody (Sigma-Aldrich, St. Louis, MO, USA) and DAPI to quantify neurite outgrowth using NeuronCyto II, a Matlab-based software package that was developed to address the “crossover” issues in quantifying neuronal morphology [25,26]. Cellular images were acquired at 4× and 20× magnification using an EVOS Cell Imaging System (Thermo Fisher Scientific, Waltham, MA, USA). The neurite outgrowth quantification algorithm of NeuronCyto II has automated features that allowed for image input, image preprocessing, segmentation, neurite tracing, quantification, batch processing to automatically quantify all acquired images under similar conditions, and results output. ImageJ was used to split each 20× image into 4 frames. Batch processing was then used to compile the length/cell measurements to calculate the total length of dendrites per image.

### 4.3. Bile Acids, Agonists, and Antagonists

The FXR activator GW 40644 [27], as well as Z-Guggulsterone, an inhibitor of FXR transactivation, were used to target FXR [30]. RG-239, a 3β-allyl semisynthetic derivative of betallinic acid, was used as a selective TGR5 activator [29]. The dual FXR/TGR5 ligand obeticholic acid (INT 747) [15] was also used. To target the liver X receptor (LXR), the LXRα agonist WAY 252623 [75] and LXRα antagonist GSK2033 [31] were used.

### 4.4. RNA Sequencing

RNA for sequencing was prepared from triplicate 10 cm tissue culture dishes plated with 300,000 NSC-34 cells per plate and treated with differentiation medium (DM), 1 μM RA, 20 μM CDCA, 20 μM DCA, and 10 μM UDCA, for 2 days. RNA samples were quantified using a Qubit 2.0 Fluorometer (Life Technologies, Carlsbad, CA, USA). Library preparation with rRNA depletion, sequencing reaction, and bioinformatics analysis was conducted at GENEWIZ, LLC. (South Plainfield, NJ, USA) using a 2 × 150 paired end (PE) configuration. Image analysis and base calling were conducted by the HiSeq Control Software (HCS). Raw sequence data generated from Illumina HiSeq were converted into fastq files and de-multiplexed using Illumina’s software. BaseSpace Sequence Hub RNA-Seq Alignment App Spliced Transcripts Alignment to a Reference (STAR) software [76] was used to align the reads to the Mus musculus UCSC mm10 (RefSeq gene annotation) genome. The RNA-Seq Differential Expression App v1.0.1 (Illumina Inc., San Diego, CA, USA) was used to perform differential expression analyses. Salmon was used to quantify the expression of genes and transcripts defined in the reference annotation. Principal component analysis was performed on all the samples. Unique gene hit counts and unique transcript hit counts were generated and used for downstream differential expression analysis. DESeq2 was used to perform the differential expression and principal components analysis [32].

### 4.5. Statistical Analysis

For the analysis of neurite length, median, rather than mean, length was used and data distributions tested for normality. For simple comparisons between two groups, a two-sample Student’s t-test was used. When comparisons among several groups are analyzed, analysis of variance (ANOVA) with an appropriate post hoc test will be used, assuming non-parametric distributions as indicated. All statistical tests will be two-sided, using Prizm 7.0, with *p* ≤ 0.05 considered statistically significant. Cell culture studies were conducted using at least 3 independent cell preparations.

## 5. Conclusions

We have shown that DCA could induce neurite outgrowth in vitro, with effects similar to the neurotrophic effects of RA, and identified a core set of genes whose expression was regulated by DCA, CDCA, and RA using RNA sequencing. TGR5 agonism was comparable to DCA stimulation, and stronger than RA. Our data suggest that bile acid signaling through TGR5 may be a novel targetable pathway to stimulate neurite outgrowth, further supporting a role for the neuroprotective and neurotrophic effects of bile acids beyond the conventional scope of liver-related disorders. Future studies will be required to further characterize the molecular mechanisms involved in the stimulation of neurite outgrowth, and to determine whether this pathway may be exploitable in other cell types and whether they may be relevant in vivo. For example, activation of TGR5 has been shown to have multiple neuroprotective effects in rodent models [77]. In vitro and in vivo TGR5 over-expression studies to increase signaling may also be informative. Further delineation of TGR5 molecular agonists and antagonists in the nervous system may shed further light on neuroprotective mechanisms, and modulation of TGR5 signaling may be a potential pathway to counter axonal loss in neurological disorders such as ALS. The native expression of TGR5 in the nervous system and endogenous bile acid agonists should enable more direct and efficient options for pharmacological intervention.

## Figures and Tables

**Figure 1 pharmaceuticals-16-00174-f001:**
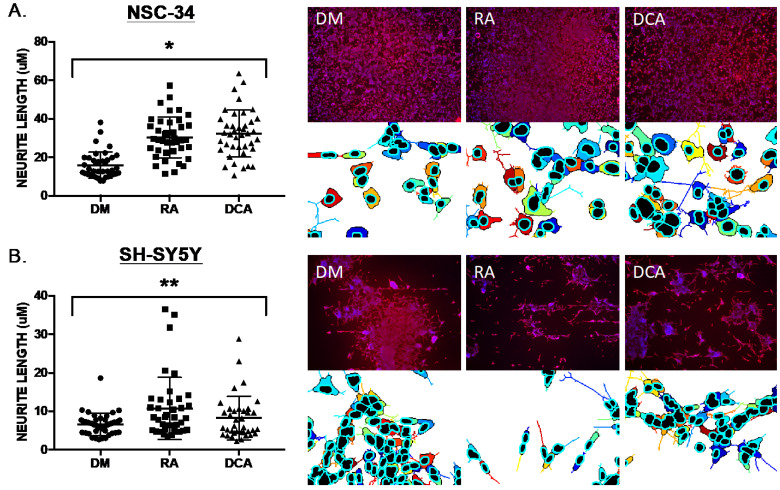
Neurite outgrowth measured as neurite length using NeuronCyto II. (**A**) NSC-34 exposed to basal medium (DM), 1 μM retinoic acid (RA), and 20 μM of the bile acid deoxycholic acid (DCA) for 6 days. Four microscopic fields (colored examples shown below the red/blue DAPI/immunostaining images) from each of ten 20X images (40 total fields) of each culture condition were processed, and neurite lengths calculated and plotted. RA and DCA resulted in increased neurite length in NSC-34 (Kruskal–Wallis statistic, * *p* < 0.0001). (**B**) SH-SY5Y cells NSC-34 exposed to basal medium (DM), 1 μM retinoic acid (RA), and 20 μM of the bile acid deoxycholic acid (DCA) for 6 days and analyzed as described in A for NSC-34 cells. RA and DCA resulted in increased neurite length in NSC-34 (Kruskal–Wallis statistic,** *p* = 0.0196).

**Figure 2 pharmaceuticals-16-00174-f002:**
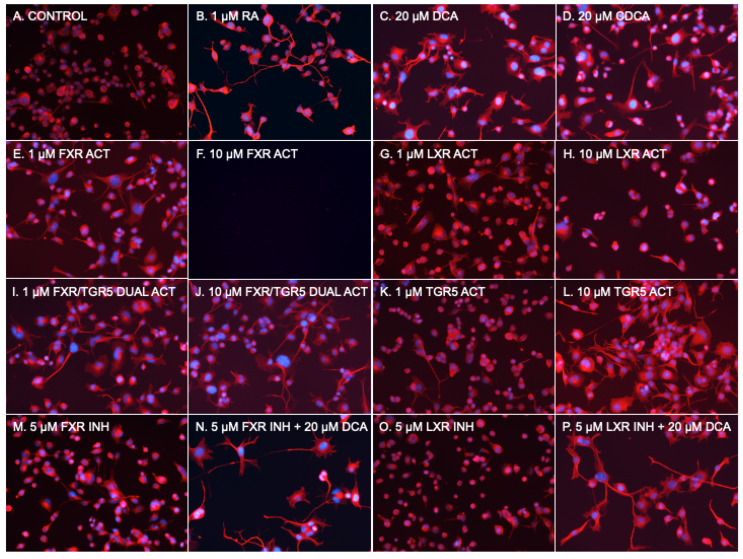
Photomicrographs of NSC-34 cells. (**A**) Basal medium. (**B**) 1 μM RA. (**C**) 20 μM DCA. (**D**) 20 μM CDCA. (**E**) 1 μM GW4064 (FXR Activator). (**F**) 10 μM GW4064 (FXR Activator). (**G**) 1 μM WAY 252623 (LXR activator). (**H**) 10 μM WAY 252623 (LXR activator). (**I**) 1 μM INT 747 (FXR/TGR5 dual activator). (**J**) 10 μM INT 747 (FXR/TGR5 dual activator). (**K**) 1 μM RG-239 (TGR5 activator). (**L**) 10 μM RG-239 (TGR5 activator). (**M**) 5 μM Z-Guggulsterone (FXR inhibitor). (**N**) 5 μM Z-Guggulsterone (FXR inhibitor) + 20 μM DCA. (**O**) 5 μM GSK 2033 (LXR inhibitor). (**P**) 5 μM GSK 2033 (LXR inhibitor) + 20 μM DCA.

**Figure 3 pharmaceuticals-16-00174-f003:**
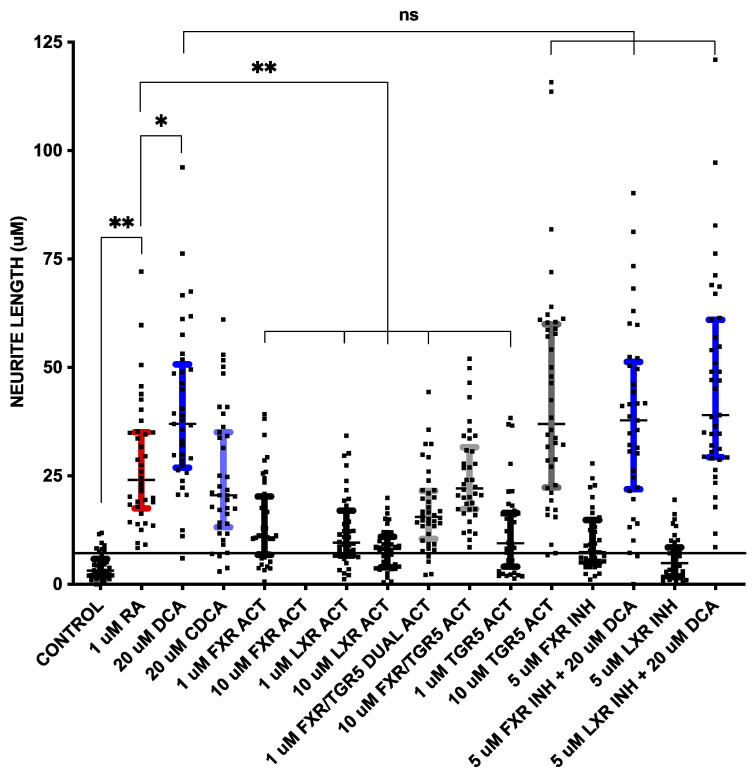
Neurite outgrowth measured as neurite length, as described in Figure 1, from cells in Figure 2. The horizontal line at ~8 μm is drawn through + 1 standard deviation of control neurite length measurements. The conditions in which 1 standard deviation did not overlap with control (i.e., above horizontal line) are colored: 1 μM RA in red, 20 μM DCA in dark blue, 20 μM CDCA in light blue, 1 μM INT 747 (FXR/TGR5 dual activator) in light gray, 10 μM INT 747 (FXR/TGR5 dual activator) in light gray, 10 μM RG-239 (TGR5 activator) in dark gray, 5 μM Z-Guggulsterone (FXR inhibitor) + 20 μM DCA in dark blue and 5 μM GSK 2033 (LXR inhibitor) + 20 μM DCA in dark blue. * *p* < 0.01, ** *p* < 0.001, ns = not significant.

**Table 1 pharmaceuticals-16-00174-t001:** Genes whose changes in expression were similar in RA and DCA or CDCA treated NCE34 cells.

Ligand	Gene	log2 Fold change	*p*-Value	Baseline *	2 days **
RA	Ikbip	5.54	0.0023	0.18	8.36
DCA	Ikbip	4.92	0.0173	0.18	5.40
CDCA	Ikbip	5.13	0.0304	0.17	5.89

RA	Kif2c	−3.93	0.0278	7.80	0.51
DCA	Kif2c	−3.98	0.0239	7.96	0.50

RA	Lgals1	−1.46	0.0029	36.16	13.17
DCA	Lgals1	−0.93	0.0471	36.93	19.34

RA	Mvb12a	2.61	0.0331	1.50	9.13
DCA	Mvb12a	2.59	0.0299	1.54	9.31

RA	Sdf4	2.38	0.0058	4.52	23.53
DCA	Sdf4	2.78	0.0001	3.70	25.37

RA	Ubald2	−6.08	0.0004	12.59	0.19
DCA	Ubald2	−6.13	0.0004	12.85	0.18

RA	Adnp	5.97	0.00061	0.18	11.12
DCA	Adnp	5.61	0.0015	0.18	8.81

RA	Marcksl1	1.25	0.0161	12.16	28.97
CDCA	Marcksl1	1.65	0.0437	11.19	35.11

RA	Mki67	−0.90	0.0137	59.13	31.76
CDCA	Mki67	−1.80	0.0125	56.19	16.09

* Expression of the control group back-calculated from base mean and log2 fold change. ** Expression of the comparison group back-calculated from base mean and log2 fold change.

**Table 2 pharmaceuticals-16-00174-t002:** Genes whose changes in expression were similar in DCA and CDCA treated NCE34 cells.

Ligand	Gene	log2 Fold change	*p*-Value	Baseline *	2 days **
DCA	Ikbip	4.92	0.0173	0.18	5.40
CDCA	Ikbip	5.13	0.0304	0.17	5.89

DCA	Gas2l3	−1.98	0.0344	12.07	3.06
CDCA	Gas2l3	−3.42	0.0349	11.72	1.09

DCA	Rps12	0.63	0.0462	59.65	92.62
CDCA	Rps12	1.66	0.0108	55.62	175.96

DCA	Tprgl	3.82	0.0354	0.50	7.00
CDCA	Tprgl	4.39	0.0276	0.45	9.49

* Expression of the control group back-calculated from base mean and log2 fold change. ** Expression of the comparison group back-calculated from base mean and log2 fold change.

## Data Availability

The datasets used and/or analyzed during the current study are available from the corresponding author on reasonable request.

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
