# Peer review of "Bile Acids Induce Neurite Outgrowth in Nsc-34 Cells via TGR5 and a Distinct Transcriptional Profile"

_pharmaceuticals, 2023, doi:10.3390/ph16020174_

Round 1

Reviewer 1 Report

This manuscript complements the literature with data on the major human bile acids as potential neurotrophic agents (signaling molecules for their two-cell receptors, farnesoid X receptor, and G protein-coupled bile acid receptor 1).

The results support the idea that bile acid signaling through TGR5 may be a targeted pathway to stimulate neurite outgrowth.

This is a good manuscript on technically correct and sums up exciting results.

It is not so well-written but presents useful methods for studying the aspects analyzed.

I recommend that the manuscript be published after Revision.

The figures are not of good quality, please consider improving them and writing the legend in order to respect the journal format (the same for the tables).

You need to improve the quality of the paper presentation by respecting the rules of the journal, using:

2.1. Subsection

2.1.1. Subsubsection

Etc.

In vivo in vitro italic in the paper, you have both versions.

References are not in the right font…

Delete the final row after references.

You need to improve rigorously the format of the paper.

Reviewer 2 Report

A growing evidence supports a neuroprotective role of bile acids in major neurodegenerative disorders.  However, its mechanism of action has been unclear.  Thus, the objective of this study is to investigate the effect of physiological levels of several of the main bile acids on neurite outgrowth.  The authors demonstrated that deoxycholic acid and chenodexoycholic acid could function as neurotrophic factors that can induce neurite outgrowth comparable to retinoic acid in mouse NSC-34 cells and human SH-SY5Y cells.  In addition, bile acids appeared to signal through TGR5 to induce neurite growth.  Furthermore, several retinoic acid and bile acid responsive genes were identified.  The authors conclude that bile acid signaling through TGR5 may be a targetable pathway to stimulate neurite outgrowth.  The manuscript is well-written and the methods sound.  I have a comment that I believe need to be addressed prior to publication of this article.

Comments:

Page 2, lines 60, “FXR also heterodimerizes with retinoid X receptors [13] whose ligands include retinoic acid, and both isoforms of LXR, LXRα (NR1H3) and LXRβ (NR1H2), form heterodimers with RXR [14], which serve as transcription factors to regulate gene expression by binding to DNA at specific sequences in the promoters and enhancers of target genes.”  Please define RXR.

Please revise “uM” throughout the text (e.g. Page 2, line 87).

Page 4, line 117, “Left Panel:”

Reviewer 3 Report

Major points:

- Authors should confirm the TGR5 involvement (and not that of FXR) with additional siRNA experiments;

- Authors should explain in more detail results of Table 1 and 2, pointing out in more detail the biological significance of the change in the expression of the reported genes;

- Readers of the present manuscript will be wondering if a similar behavior, in term of neurite outgrow and change in the transcriptional profile, occurs also in models of pathological conditions (NSC-34 cells overexpressing TDP-43, or ipsc-derived motor neurons from als patients). This aspect should be at least discussed.

Minor points

- The work by N. Cashaman "Cashman NR, Durham HD, Blusztajn JK, Oda K, Tabira T, Shaw IT, Dahrouge S, Antel JP. Neuroblastoma x spinal cord (NSC) hybrid cell lines resemble developing motor neurons. Dev Dyn. 1992 Jul;194(3):209-21." should be cited when NSC-34 cells are described;

- The statistical analysis of Fig. 3 should be introduced in the histogram.

Reviewer 4 Report

The manuscript entitled "Bile acids induce neurite outgrowth in NSC-34 cells via TGR5 2 and a distinct transcriptional profile" is well written and described.

The topic is of current interest but I believe the conclusion is underdeveloped. The authors should elaborate on the conclusion part and provide a future direction based on the outcome.

Also, recommended to include the following relevant references

doi: 10.1096/fj.201600275R

https://doi.org/10.3390/ijms21175982

Round 2

Reviewer 1 Report

The authors made the changes required.

The manuscript can be accepted in its present form.

Reviewer 3 Report

Authors adequately addressed my issues